# The Membrane Electrical Potential and Intracellular pH as Factors Influencing Intracellular Ascorbate Concentration and Their Role in Cancer Treatment

**DOI:** 10.3390/cells10112964

**Published:** 2021-10-30

**Authors:** Mateusz Gąbka, Paulina Dałek, Magdalena Przybyło, Daniel Gackowski, Ryszard Oliński, Marek Langner

**Affiliations:** 1Faculty of Fundamental Problem of Technology, Wroclaw University of Science and Technology, Wybrzeże Stanisława Wyspiańskiego 17, 50-370 Wrocław, Poland; 236501@student.pwr.edu.pl; 2Department of Biomedical Engineering, Wrocław University of Science and Technology, Wybrzeże Stanisława Wyspiańskiego 17, 50-370 Wrocław, Poland; magdalena.przybylo@pwr.edu.pl (M.P.); marek.langner@pwr.edu.pl (M.L.); 3Lipid Systems Sp. z o.o., ul. Krzemieniecka 48c, 54-613 Wrocław, Poland; 4Department of Clinical Biochemistry, Faculty of Pharmacy, Collegium Medicum in Bydgoszcz, Nicolaus Copernicus University in Toruń, Karłowicza 24, 85-092 Bydgoszcz, Poland; danielg@cm.umk.pl (D.G.); ryszardo@cm.umk.pl (R.O.)

**Keywords:** vitamin C, membrane electrical potential, cancer treatment, ferroptosis, ascorbate efficacy

## Abstract

Ascorbate is an important element of a variety of cellular processes including the control of reactive oxygen species levels. Since reactive oxygen species are implicated as a key factor in tumorigenesis and antitumor therapy, the injection of a large amount of ascorbate is considered beneficial in cancer therapy. Recent studies have shown that ascorbate can cross the plasma membrane through passive diffusion. In contrast to absorption by active transport, which is facilitated by transport proteins (SVCT1 and SVCT2). The passive diffusion of a weak acid across membranes depends on the electrostatic potential and the pH gradients. This has been used to construct a new theoretical model capable of providing steady-state ascorbate concentration in the intracellular space and evaluating the time needed to reach it. The main conclusion of the analysis is that the steady-state intracellular ascorbate concentration weakly depends on its serum concentration but requires days of exposure to saturate. Based on these findings, it can be hypothesized that extended oral ascorbate delivery is possibly more effective than a short intravenous infusion of high ascorbate quantities.

## 1. Introduction

Ascorbate is a compound indispensable for maintaining body homeostasis. It is necessary for a variety of metabolic processes ranging from the maintenance of the extracellular matrix to ensuring the effective flow of information between the cell genome and the proteome [1,2]. Its deficiency leads to pathologies ranging from mild symptoms to death, depending on the depth and duration of the deficiency [3]. The distribution of ascorbate in the human body correlates directly with the metabolic activity of tissues and cells, namely, the higher the local metabolic activity, the more ascorbate is required [4]. The ascorbate homeostasis in extracellular fluids and its complex distribution inside cells are maintained by the combination of passive and active transports. The active ascorbate influx to the cytoplasm is facilitated by two dedicated transporters, SVCT1 and SVCT2 [5]. The importance of the intracellular concentration of ascorbate, which is predominantly controlled by SVCT2 transporters, has been demonstrated using knock-out mice. While mice without SVCT1 were able to survive, those without SVCT2 die at birth [4]. This shows that the distribution of ascorbate among cells is fundamental for the survival of the organism. As described elsewhere, the level of intracellular ascorbate is maintained by the balance of two fluxes, both powered by genetically controlled active transport [6], where the passive transport dissipates the ascorbate concentration gradient generated by the active transport across the plasma membrane. Being a critical element of body homeostasis, the level of ascorbate in extracellular fluids is tightly controlled and the excess is rapidly excreted [7]. Ascorbate is generally considered nontoxic, however, when added to tissue cultures in concentrations of a few mM causes the death of cancer cells. These experiments indicate that cancerous cells are sensitive to elevated concentrations of ascorbate [6]. Based on such data, the treatment of various cancers has been proposed in which a large amount of ascorbate is administered intravenously [8,9,10]. The therapy protocol is directly translated from classical pharmacology, where the concentration of an active substance in the blood is sufficient to assess its concentration in tissues of interest. This critical assumption might not hold for ascorbate due to the existence of specific membrane transporters and physiological control of its concentration in serum [8]. Consequently, the effort to elevate the serum concentration of ascorbate may not be required to achieve the therapeutic effect, especially when the half-time of the elevated concentration in serum is shorter than an hour [9]. Current protocols of the procedure are the result of a trial and error approach aiming at reaching serum concentrations when ascorbate is toxic to cancer cells, i.e., a few mM [10]. The extended in-time infusion of large quantities of ascorbate is inconvenient and its outcome is uncertain, due mainly to the pure understanding of the molecular mechanisms responsible for the selectivity of the treatment toward cancer cells. It has been postulated that a high level of ascorbate leads to an increased quantity of labile iron in the cytoplasm causing a massive rise of reactive oxygen species and subsequent cell death (ferroptosis) [11,12]. Similar effects accompany chemotherapy and are required for cancer cell damage [13]. Therefore, the ability to generate reactive oxygen species positions ascorbate as a potentially useful compound in cancer therapy.

In the paper, the new model of ascorbate intake by cells, which accounts for the passive diffusion, is presented. The important consequence of ascorbate ability to penetrate the lipid bilayer is that the electrical potential of the plasma membrane and the local pH should be included in the evaluation of the steady-state intracellular ascorbate concentration [14]. This opens the possibility to use the plasma membrane electrical potential difference and/or local pH as a selectivity tool.

For the sake of simplicity, the ascorbate-dependent single mechanism causing cell death was selected. Specifically, the sufficiently high concentration of ascorbate inside a cell may reduce ferric ions to highly toxic labile ferrous ions, which will likely lead to cell death by unspecific destruction of vital cellular components (ferroptosis [15,16]). The ascorbate-driven release of ferrous ions from its intracellular protein stores cannot be easily controlled by the cell, contrary to exogenous drugs [17]. This major difference between exogenous drugs and ascorbate in maintaining an elevated intracellular concentration is due to the fact that, while there are no specific transporters for an exogenous drug, ascorbate is actively accumulated within cells by dedicated membrane proteins (SVCT1 and SVCT 2) [18]. In contrast, the exogenous drug can be actively removed from the cytoplasm by ABC transporters leading to drug resistance [19,20]. Since all known membrane transporters are designed to condense ascorbate in the cytoplasm, therefore, the cell does not have effective molecular defenses against the elevated level of ascorbate in the cytoplasm. The only ways to reduce the ascorbate concentration inside the cell is its passive flow across the plasma membrane or its oxidation to DHA, which subsequently diffuses out of the cell through GLUT channels or is rapidly reduced back to ascorbate by metabolic processes inside the cell. Therefore, the oxidation of ascorbate will not affect significantly its cytoplasmic concentration [6]. The excessive level of ascorbate in the cytosol may trigger ferroptosis, leading to cell death [21]. The intracellular level of ascorbate in a specific cell depends, in addition to metabolic consumption, on a number of ascorbate transporters and electrical and pH gradients across its plasma membrane [8]. Due to the differences in plasma membrane electric potential and local pH, ascorbate will rise in cancer cells, potentially leading to their ferroptosis, whereas healthy cells will be little affected [8,22]. In the paper, we present a quantitative analysis aiming at determining the dependence of intracellular ascorbate concentration at the steady-state on its concentration in serum as well as the time needed to reach the steady-state level for healthy and malignant cells.

The model, contrary to previous approaches, is the first attempt to evaluate quantitatively the supportive treatment of cancer patients using ascorbate [19]. The model is founded on two new findings, that ascorbate diffuses passively across the lipid bilayer and that ascorbate-triggered ferroptosis is the mechanism affecting targeted cells. The main finding of the analysis is that, due to the active transport, the serum concentration of ascorbate is not a good indication of the treatment efficacy. The intracellular ascorbate concentration should be measured instead. This makes the theoretical model, after validation, an attractive tool for planning the ascorbate-based therapy.

## 2. Materials and Methods

All calculations have been performed using dedicated Python (v3.8.10) scripts with NumPy v1.17.4 numerical library [20]. Model’s equations were implemented as separate functions, where all parameters were expressed in SI units. All calculations were performed on NumPy 1D-arrays using a float 64 number format. Data have been plotted in a 3D and 2D projection on 50- or 200- point arrays using Python graphical library Matplotlib v3.3.4. [23].

## 3. Results

In the model, the human body with cancer has been reduced to two different populations of cells immersed in a single aqueous compartment. Selected parameters of cancer and healthy cells are listed in Table 1.

At the first approximation, the differences between the local tumor microenvironment and extracellular fluids in healthy tissues have been neglected [16,30,31]. These assumptions are not sufficient when ascorbate distribution within tumors is analyzed. However, this simplification will not mitigate the derived conclusions, since the lower pH and possible enhanced level of iron ions inside tumors will rather enhance the effect [32,33,34,35]. It was also assumed that the extracellular ascorbate concentration is invariant in time during treatment due to the adequate supplementation [7,36]. The therapeutic efficacy of ascorbate depends predominantly on its intracellular concentration, which may elevate the concentration of labile iron ions to the level when radical oxygen species are excessively produced, leading to cell death [9,13].

The ascorbate treatment is simulated by the constant ascorbate concentration in the extracellular fluid compartment $C_{ext}$, which serves as an infinite, continuously resupplied, reservoir for both healthy and cancer cells. It has been assumed that the extracellular ascorbate concentration depends on the balance between the rate of delivery (infusion or supplementation) ($J_{delivery}$), the intake by cells ($J_{cell})\;$and the secretion $(J_{sec})\;$(Equation (1)).
(1)$$\frac{dC_{ext}}{dt}=J_{delivery}\left(t\right)+J_{cell}\left(t\right)+J_{sec}\left(t\right)$$

During the therapy, the $J_{delivery}\left(t\right)$ is maintained by the application protocol. The $J_{cell}\left(t\right)$ depends on the temporal metabolic activity, therefore the $J_{sec}\left(t\right)$ will depend on the ascorbate quantity delivered to the body, metabolic consumption and the quantity needed for the restoration of intracellular pools.

Any therapy aimed at the eradicating of cancer cells relies on the differences between cancer and normal cells regarding their physiology and/or metabolic state, so the specificity of the therapy can be achieved. The ascorbate-based cancer treatment is based on the difference between healthy and cancer cells with respect to their sensitivity to extracellular ascorbate concentration. All previous ascorbate-based therapies were founded on the assumption that ascorbate affects a specific molecular process in cancer cells in a concentration-dependent meaner, in line with classical pharmacological thinking [9,19]. However, the application of ascorbate as a pharmacological agent is different. Depending on its intracellular concentration, its function may vary from performing the cofactor and antioxidant functions, being a necessary element of critical metabolic processes, to causing cell death by triggering the excessive production of reactive oxygen species (ROS) by causing the rise of the labile iron pool in the cytoplasm [19,30,31]. The triggering of the transformation takes place at a specific ascorbate concentration inside the cell [16,37]. This functional duality can be utilized in cancer treatment only if there is a selectivity mechanism affecting the acquisition of ascorbate by cells. In the model, it was assumed that the cellular selectivity in ascorbate-based cancer treatment is achieved thanks to differences in ascorbate intake by different cells. Even under physiological conditions, the intracellular ascorbate concentrations may vary from 0.05 to 10 mM [4]. The temporal intracellular ascorbate concentration is a result of the balance between active transport, metabolism and leakage, the latest being facilitated by passive diffusion across the lipid bilayer [8]. The intracellular steady-state ascorbate concentration depends on cell properties such as cytoplasmic pH, electric membrane potential, number of SVCT2 transporters present in the plasma membrane and metabolic activity. Table 1 presents differences between healthy and cancer cells, which may affect intracellular ascorbate concentration and may serve as a source of therapeutic selectivity [8,38].

Figure 1 shows schematically molecular mechanisms leading to cancer cell destruction induced by elevated intracellular ascorbate concentration. The presented analysis has been performed for imaginary healthy and cancer cells immersed in unified extracellular fluid. The differences in extracellular pH between a specific healthy tissue and tumor had not been accounted for. The two types of selected imaginary cells, differing with respect to their plasma membrane electrical potentials and intracellular pH, have been compared. Cells were selected in such a way so the possibility to use the electrical plasma membrane as a selectivity parameter. The model can be easily adopted to a specific cancer cell type by changing the relevant parameters.

The intracellular concentration of ascorbate depends on the temporal cell metabolic activity and fluxes acres the plasma membrane (Equation (2)).
(2)$$\frac{dC_{i}}{dt}=J_{\mathrm{active}}+J_{\mathrm{passive}}+J_{\mathrm{metabolism}}$$

It has been assumed that at elevated ascorbate extracellular concentrations (higher than 0.1 mM) and with extracellular Na^+^ concentration unchanged, the SVCT transporters will work with their maximum velocity (*V*_max_) [38]. Additionally, it has been assumed that during the ascorbate delivery, cells do not respond on the proteomic level, i.e., the number of ascorbate transporters (N_SVCT2_) does not change. Therefore:(3)$$J_{\mathrm{active}}=constant=N_{SVCT2}V_{\max}$$

The passive ascorbate transport depends, in addition to the membrane electric potential, on concentration gradients of its neutral and ionized forms, which in turn depend on local pH [14,39]. The surface-area normalized net flux of monoprotic acid across the plasma membrane (J_passive_) is represented by Equation (4) [14]. In the equation, *P_n_* and *P_i_* represent the permeabilities of the neutral and ionized forms of ascorbate. *fu_d_*, *C_i_*, *fu_r_* and *C_e_* represent the unbound fraction and total ascorbate concentrations in the extracellular (*e*) and intracellular (*i*) aqueous compartments. *H_e_* and *H_i_* represent the fraction of ionized ascorbate in the extracellular and intracellular aqueous compartments. *K_e_* and *K_i_* represent the respective neutral (protonated) fraction of ascorbate [14].
(4)$$J_{\mathrm{passive}}=P_{n}\left(fu_{e}K_{e}C_{e}-fu_{i}K_{i}C_{i}\right)-\frac{vP_{i}\left(fu_{i}H_{i}C_{i}-e^{-v}fu_{e}H_{e}C_{e}\right)}{1-e^{-v}}$$
where
(5)$$v=\frac{zF\Delta \varphi }{RT}$$

*F*, Δφ, *R* and *T* stand for Faraday constant, membrane potential difference, gas constant and temperature, respectively.

Equation (4) can be rearranged to the following form [14]:(6)$$J_{\mathrm{passive}}=C_{e}\alpha_{e}-C_{i}\alpha_{i}$$
where $\alpha_{e}$ end $\alpha_{i}$ are coefficients depending on membrane electrical potential difference and local pH:(7)$$\alpha_{e}=\frac{P_{n}10^{pK_{a}}\left(1-e^{-v}\right)+P_{i}10^{pH_{e}}ve^{-v}}{\left(1-e^{-v}\right)\left(10^{pH_{e}}+10^{pK_{a}}\right)}$$
(8)$$\alpha_{i}=\frac{P_{n}10^{pK_{a}}\left(1-e^{-v}\right)+P_{i}10^{pH_{i}}v}{\left(1-e^{-v}\right)\left(10^{pH_{i}}+10^{pK_{a}}\right)}$$

Finally, the Equation (5) can be presented in the form:(9)$$J_{\mathrm{passive}}=\frac{A_{\mathrm{cell}}}{V_{\mathrm{cell}}}\left(C_{e}\alpha_{e}-C_{i}\alpha_{i}\right)$$

The net ascorbate depletion due to metabolic processes can be estimated as follows:(10)$$J_{\mathrm{metabolism}}=-m_{\mathrm{protein}}\frac{v_{\mathrm{met}}}{V_{\mathrm{cell}}}$$

The metabolic activity $v_{\mathrm{met}}$ as evaluated by Marin–Hernandez for glucose is used as the measure of the ascorbate consumption [4,40]. The approximation is based on two observations; that metabolic activity of cancer cells exceeds that of normal cells and that the ascorbate required for sick patients to reach its homeostatic level is much higher than for healthy persons [41,42].

It follows that the ascorbate intracellular concentration change can be formulated by the following equation:(11)$$\frac{dC_{i}}{dt}=N_{\mathrm{SVCT}2}V_{\max}+\frac{A_{\mathrm{cell}}}{V_{\mathrm{cell}}}\left(C_{e}\alpha_{e}-C_{i}\alpha_{i}\right)-\frac{m_{\mathrm{protein}}v_{\mathrm{met}}}{V_{\mathrm{cell}}}$$

Since the therapy based on the ascorbate delivery is performed for an extended period of time, it has been assumed that the steady-state intracellular ascorbate concentration will be reached. Consequently, the intracellular ascorbate concentration can be calculated under the condition of the balance of all fluxes:(12)$$\frac{dC_{i}}{dt}=0$$

Consequently,
(13)$$N_{\mathrm{SVCT}2}V_{\max}+\frac{A_{\mathrm{cell}}}{V_{\mathrm{cell}}}\left(C_{e}\alpha_{e}-C_{i}\alpha_{i}\right)-\frac{m_{\mathrm{protein}}v_{\mathrm{met}}}{V_{\mathrm{cell}}}=0$$

Since the ascorbate concentration in the serum and interstitial spaces is maintained, thanks to continuous supply, therefore the steady-state intracellular ascorbate concentration should satisfy the following equation:(14)$$C_{i,\mathrm{ss}}=\frac{V_{\mathrm{cell}}N_{\mathrm{SVCT}2}V_{\max}+A_{\mathrm{cell}}C_{e}\alpha_{e}-m_{\mathrm{protein}}v_{\mathrm{met}}}{\alpha_{i}A_{\mathrm{cell}}}$$

Equation (11) correlates the intracellular ascorbate concentration with its concentration in serum, which accounts for relevant cellular parameters. In order for the therapy to be effective, the intracellular ascorbate concentration should exceed the concentration required to release iron from its binding site triggering the ferroptosis [17]. The difference in the intracellular ascorbate concentrations between cancer and healthy cells opens the possibility for designing the ascorbate-based cancer treatment with a specific dosage and duration [43]. Equation (11) can be used to calculate the steady-state intracellular concentration of ascorbate as a function of its extracellular concentration (dosage) for a cell having specific electrical membrane potential and cytoplasmic pH.

The other important therapy parameter is its duration. For that reason, the time required to reach the desired steady-state intracellular concentration should be estimated. The dependence of the intracellular ascorbate concentration on time can be expressed in the following simplified form:(15)$$\frac{dC_{i}}{dt}=-aC_{i}-bt\;$$
where
(16)$$a=\frac{A_{\mathrm{cell}}}{V_{\mathrm{cell}}}\alpha_{i}\;\mathrm{and}\;b=-\left(N_{\mathrm{SVCT}2}V_{\max}+\frac{A_{\mathrm{cell}}}{V_{\mathrm{cell}}}C_{e}\alpha_{e}-\frac{m_{\mathrm{protein}}v_{\mathrm{met}}}{V_{\mathrm{cell}}}\right)$$

The Equation (12) has a physically meaningful solution:(17)$$C_{i}\left(t\right)=ce^{-at}-\frac{b}{a}$$
where c is a constant related to the initial intracellular ascorbate concentration, which could be evaluated using boundary conditions:(18)$$C_{i}\left(t=0\right)=c-\frac{b}{a}\Rightarrow c=C_{i}\left(t=0\right)+\frac{b}{a}$$
whereas at steady-state limit
(19)$$\frac{b}{a}=-C_{i,\mathrm{ss}}$$

Consequently,
(20)$$C_{i}\left(t\right)=\left(C_{i,0}-C_{i,\mathrm{ss}}\right)e^{-\frac{A_{\mathrm{cell}}}{V_{\mathrm{cell}}}\alpha_{i}t}+C_{i,\mathrm{ss}}$$

The time needed when the intracellular concentration reaches the value equals half of the steady-state intracellular ascorbate concentration, which can be calculated from the following equation: (21)$$\tau_{1/2}=-\frac{V_{\mathrm{cell}}}{A_{\mathrm{cell}}\alpha_{i}}\ln\left(\frac{-\frac{1}{2}C_{i,\mathrm{ss}}}{C_{i,0}-C_{i,\mathrm{ss}}}\right)$$

The main objective of the presented analysis is to design the effective ascorbate-based cancer treatment using the mathematical modelling.

To describe the ascorbate-based therapy, two ranges of ascorbate concentrations in serum have been defined. The first range corresponds to the physiological level, which can be maintained by providing ascorbate with foodstuff (<0.1 mM) [40]. Interestingly, this limit coincides with the SVCT2 saturation level [38]. Above that level, the SVCT2 transporters are saturated and intracellular ascorbate concentration can reach levels harmful to susceptible cells. It is generally accepted that the effective therapeutic level can be achieved only by intravenous infusion of a large quantity of ascorbate [9]. One of the objectives of the simulation is to determine whether therapeutic concentrations can be reached by prolonged oral supplementation with traditional (<0.250 mM [7]) or liposomal (<0.5 mM [6]) preparations.

All quantitative measures derived during the calculations describe a simplified situation, where two hypothetical cell populations representing healthy and cancer cells are considered (Table 1). When healthy cells plasma membrane electrical potential was assumed to be equal to −100 mV (negative inside the cell) and cytoplasmic pH = 7.0 the electrical potential difference across the plasma membrane of cancer cells was assumed to be equal to −10 mV and cytoplasmic pH = 7.5.

The concentration of an exogenous active substance in serum is traditionally used as a therapeutic parameter, which is correlated with the flux of an exogenous active substance into cells. However, in the case of endogenous substances such as ascorbate, the descriptor is not very useful. The flow of ascorbate mono-ions towards the cytoplasm is facilitated by SVCT transporters and equals the V_max_ if the serum ascorbate concentration exceeds 0.1 mM and remains unchanged at higher values [44]. The intracellular ascorbate concentration sufficient to trigger ferroptosis depends exclusively on the intrinsic properties of these transporters and elevation of serum concentration of ascorbate above 0.1 mM limit will not have a significant effect. To trigger the therapeutic effect, i.e., ferroptosis, a certain intracellular ascorbate concentration is required. We arbitrarily assumed that the concentration should be equal to 20 mM, as this value exceeds twice the highest physiological intracellular value. The intracellular concentration of ascorbate in neurons reaches the level of 10 mM ([6] and citations therein). It has been assumed that the metabolism of dehydroascorbic acid (DHA) can be disregarded as a significant factor in triggering ferroptosis because its concentration is two orders of magnitude lower than that of ascorbate. In addition, when inside cells, it is rapidly reduced back to ascorbate by glutathione peroxidase activity and NADPH [4,9].

First, the effect of extracellular ascorbate concentration and membrane electrical potential on the steady-state intracellular ascorbate concentration, both for healthly and cancer cells, has been evaluated (Figure 2). The calculated steady-state intracellular ascorbate concentration in the whole range of parameters used is significantly higher in cancer cells than in healthy cells. Interestingly, in both cell types, the steady-state intracellular ascorbate concentration only weakly depends on its extracellular concentration. This is due to the fact that the intracellular ascorbate level is facilitated by SVCT transporters. The effect of electrical membrane potential is larger and affects cancer cells to a much greater extent, regardless of the value of the extracellular ascorbate concentration. The steady-state intracellular ascorbate concentration in cancer cells is significantly higher than the assumed 20 mM, the arbitrarily selected concentration when ferroptosis is triggered. According to the model, the ferroptosis occurs even at ascorbate serum concentration in ranges possible to achieve with oral ascorbate delivery (0.1–0.5 mM [6,7]). 

The results presented in Figure 2 show that steady-state intracellular ascorbate in cancer cells reaches values substantially higher than those of healthy cells, indicating a high level of selectivity of treatment. In addition, it seems that there is no difference between oral and intravenous delivery routes since there is little difference between steady-state ascorbate intracellular concentrations when its concentration in serum equals 0.2, 0.5 or 1 mM. Figure 3 shows also that the steady-state intracellular ascorbate concentration depends on the intracellular pH. This is due to the ascorbate protonation level inside the cell that affects dramatically its passive leakage [25]. The effect of intracellular pH for cells differing with the plasma membrane electrical potential is presented for three relevant extracellular ascorbate concentrations, i.e., 0.2, 0.5 and 1 mM, in Figure 3.

Figure 3 shows the effect of intracellular pH (pH_i_) and extracellular pH (pH_e_) on the intracellular steady-state ascorbate concentration, when its extracellular concentration equals 0.2, 0.5 and 1 mM and plasma membrane electrical potential equals −100 and −10 mV for healthy and cancer cells, respectively. The numerical analysis shows that the intracellular steady-state ascorbate concentration is very sensitive to pH differences between extracellular and intracellular fluids. The dependence is further enhanced by plasma membrane electrical potential difference, confirming the cancer cell susceptibility to the elevated ascorbate concentrations. 

Figure 4 shows how the two cellular properties (electrical potential difference across the plasma membrane and intracellular pH) affect the steady-state intracellular ascorbate concentration. For each extracellular ascorbate concentration, intracellular steady-state values for healthy and cancer cells are indicated.

The three graphs in Figure 4 show that the steady-state intracellular ascorbate concentration only weakly depends on its extracellular concentration and that the intracellular pH and membrane electrical potential difference may serve as selectivity parameters in ascorbate-based treatment. The other factor affecting the steady-state intracellular ascorbate concentration is the number of SVCT transporters. The number of SVCT transporters is epigenetically controlled and may influence the effect of the treatment, especially when the treatment is extended in time [45]. In addition, genetic variants of transporters may influence the outcome of the treatment [46,47].

Figure 5 shows that the epigenetic response of cancer cells is much stronger than healthy cells with respect to their ability to modulate their intracellular ascorbate concentration. It has been demonstrated above that steady-state intracellular ascorbate concentration in cancer and healthy cells can be used to design effective ascorbate-based treatment of cancer, hence cancer cell response is significantly stronger than healthy cells when exposed to the elevated ascorbate concentrations. An additional critical parameter of such treatment is the time required to reach the steady-state intracellular ascorbate concentrations. 

Figure 6 shows the time course of the intracellular ascorbate concentration for healthy and cancerous cells for three extracellular ascorbate concentrations. The ascorbate intracellular concentration in healthy cells does not exceed the assumed ferroptosis threshold value (20 mM). In the cancer cells after 10 h exposure, the intracellular ascorbate concentration approaches the tipping point at which ferroptosis is triggered. Three traces for 0.2, 0.5 and 1 mM are presented to show that in the case of ascorbate treatment, the serum concentration is not so important, but the time of exposure is critical. In practice, there are three possible ways of ascorbate supplementation; intravenous infusion [9], oral supplementation with ascorbate alone or encapsulated inside liposomes [6,7]. Presented results show that the ascorbate concentration in cancer cells can reach the level when ferroptosis is triggered for all investigated extracellular concentrations (from 0.2 to 1 mM).

In healthy cells, the 20 mM intracellular concentration cannot be reached even when the treatment is continued for an extended period of time. Since the number of SVCT transporters in the plasma membrane is a parameter which can be controlled by adjustment of epigenetic processes by cells themselves and because its influence on the intracellular ascorbate concentration is significant (Figure 5), its effect on the time needed to reach the ascorbate steady-state intracellular concentration needs to be evaluated. Figure 7 shows how the number of SVCT2 transporters influences the steady-state intracellular concentration of ascorbate in healthy and cancer cells. The dependences of the intracellular ascorbate concentrations on time were plotted for two cases when the number of SVCT2 transporters differs by a factor of four. In both cases, the concentration of ascorbate increases together with the number of SVCT2 transporters. However, for the cancer cells, the rise is much more significant. Again, the effect of extracellular ascorbate concentration on both the level and time traces of intracellular concentration is insignificant, showing that not the serum ascorbate concentration but its persistence in time is a critical factor.

Obtained results show the therapeutic effect can be achieved when ascorbate serum concentration is facilitated by supplementation using all available delivery routes as long as the exposure time is sufficiently long (days). The complete characterization of the procedure requires the evaluation of the therapeutic index as an indication of the procedure safety. Based on the proposed mechanism (ferroptosis) of ascorbate action, two toxicities can be expected; the induction of ferroptosis in healthy cells and systemic toxicity caused by dissociation of iron from transport proteins in serum. Both potential sources of toxicity await future quantitative evaluation.

## 4. Discussion

In classical pharmacology, the active compound is of exogenous origin and is designed to reach and affect specific molecular targets [8]. Typically, the internalization of such a compound by a cell is not facilitated by membrane proteins, but rather it diffuses across the lipid bilayer [14]. The efficacy of passive diffusion depends predominantly on the concentration of the active compound in the serum. This is because passive diffusion necessitates a concentration gradient across the cellular plasma membrane. This very unspecific process is a source of systemic toxicity and the selectivity towards targeted cells is based on the preferential affinity towards a carefully selected molecular entity. However, the serum concentration of the drug can be elevated only to the level when it becomes toxic [8]. Therefore, the unspecific toxicity is an intrinsic property of all low-molecular-weight exogenous compounds intended for use as pharmacologically active substances. There is yet another problem with the classical pharmacological approach, passive diffusion of the substance can be counter by cells with active outflow transport (ABC transporters [48,49] leading to the development of the resistance to the drug used in the treatment.

Ascorbate is a compound involved in numerous metabolic processes, so it may be considered a necessary element of intracellular homeostasis by serving as an electron donor or acceptor in various cellular compartments [4,40]. The critical function of ascorbate in cellular homeostasis necessitates its tight and function-driven control, both on the cellular and physiological levels. The spatial variation of ascorbate within the body is ensured by the difference in the expression of dedicated ascorbate transporters and recently discovered passive transport [8,25]. The capacity to adjust the local ascorbate concentration within the body, combined with effective elimination mechanisms, predispose ascorbate to be used as a therapeutic agent.

Numerous experiments using cell culture show that ascorbate at concentrations above a few mM result in cancer cell death ([9] and citations therein). This observation provides an indication for cancer treatment based on the extended intravenous infusion of high quantities of ascorbate [50,51]. Extended exposure is justified by the fact that any systemic excess of ascorbate is effectively eliminated from the circulation within an hour [7]. The ascorbate-based treatment relies on the ascorbate concentration-dependent activity; at low concentrations, it is an effective antioxidant and a necessary enzymatic cofactor, whereas at higher concentrations it causes a dramatic free radical rise due to the reduction of the ferric ion to a highly active ferrous ion [17]. The ascorbate metabolic functions as an electron donor or acceptor make ascorbate a good candidate for pharmacological applications, where its intracellular concentration can be used as a therapeutic target. Despite the limited understanding of the ascorbate role as a homeostatic agent, attempts have been made to design a therapeutic procedure. Widely used, albeit very unreliably, intravenous infusions have been tried [8,52].

Since the molecular mechanism leading to ascorbate selective killing of cancer cells has not been unequivocally determined, the critical parameters of the intravenous infusion cannot be exactly established [19]. This is perhaps the reason why the outcomes of various trials have been inconclusive [10,19,36,51].

The model presented in the paper shows that the ascorbate concentration in serum is not a good parameter for designing an effective therapy. This is because passive diffusion is not a mechanism responsible for the ascorbate entering the cellular cytoplasm, as is the case for all exogenous pharmaceuticals [14]. Instead, ascorbate monoanion is actively accumulated inside the cell by dedicated plasma membrane transporters. When transporters are saturated, at about 0.1 mM ascorbate serum concentration [38], further elevation of its concentration is not productive, on the contrary, when the toxic level is reached, the systemic toxicity may occur.

Considering the recent discovery of ascorbate passive diffusion across lipid bilayers and assuming that ferroptosis is the main mechanism of ascorbate therapeutic activity, a new model has been proposed. The model shows that the serum ascorbate concentration is not of critical importance but rather the duration of the treatment is. It has been determined that the treatment duration should be in the range of days rather than hours. The model predicts that the delivery of ascorbate through the oral route for a sufficiently long time should give better therapeutic results than relatively short (few hours) intravenous infusion of its large quantities. The experimental validation of the model prediction will offer a very convenient and potentially effective supportive treatment of a variety of cancers.

It has been demonstrated that pharmacological concentration of ascorbate in vivo cannot be reached by oral delivery of its classical form and that such delivery does not produce a sufficient level of hydrogen peroxide in extracellular fluid in vivo [53]. However, neither weekly IV administration for two hours of a pharmacological dose of ascorbate to prostate cancer patients induced tumor regression [54]. The proposed model shows the relevance of ascorbate as a therapeutic agent, the effectiveness of which will be dependent on its plasma concentration, capacity for uptake by tumor cells and modulation of anti-cancer activity, i.e., induced oxidative stress and ability to modulate epigenetic status of DNA. I.V. administration is invasive, exhausting for the patient and has been administrated, almost exclusively, to oncology patients with advanced cancers. However, it is possible that the limitations of the IV (insufficient treatment time) and oral administration of its classical form may be, at least partially, overcome by oral administration of its liposomal form [6].

The effect of ascorbate on cancer cells in a solid tumor is not as straightforward as in the case of the isolated malignancy (presented model). The tumor environment will surely differ from extracellular fluid with respect to the importance for SVCT transporter sodium concentration, reduced local pH and hypoxia. They will change the dependence of intracellular ascorbate concentration on its local availability. The effect will not only change the steady-state conditions but also the dynamics of reaching therapeutic levels by the intracellular ascorbate concentration in cancerous cells.

## 5. Conclusions

The concentration of ascorbate inside a cell is a result of three processes, metabolic degradation, active transport facilitated by dedicated transporters (SVCT1 and SVCT2) and the passive diffusion across the lipid bilayer of the plasma membrane. The dependence of the passive transport on ascorbate protonation and plasma membrane potential difference can be used for selective targeting of ascorbate to cancer cells. Specifically, most cells constituting the human body are highly differentiated. Such cells are characterized by, among other things, high negativity inside the plasma membrane electrical potential (reaching −100 mV). Cancer cells are different, as their electrical plasma membrane potential is usually low and does not exceed −10 mV. This difference can be used to elevate the ascorbate concentration in the cytoplasm of cancer cells without affecting healthy cells. This effect may justify the application of ascorbate as the supportive treatment in cancer therapy characterized by none or low systemic toxicity. Simulations presented in the paper show that such treatment to be effective requires maintenance sufficient, albeit not necessary excessively high, ascorbate concentration in plasma and interstitial fluids. However, due to the limited capacity of the active transport, the ascorbate exposure should be extended in time for days rather than hours.

## Figures and Tables

**Figure 1 cells-10-02964-f001:**
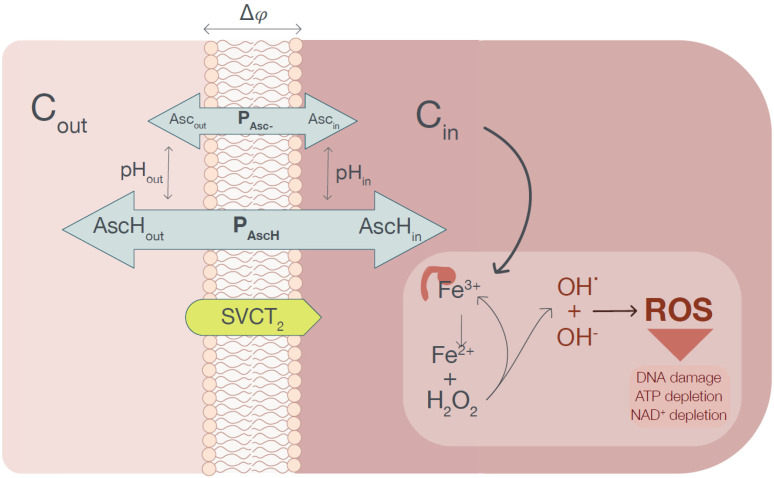
The graphical representation of the conceptual model used for the calculation of the steady-state intracellular ascorbate concentration. The ascorbate is transferred from the extracellular phase into a cell cytoplasm by an SVCT transporter. The generated ascorbate concentration gradient powers the passive ascorbate transport, which depends on the plasma membrane electrical potential Δφ and pH in compartments separated by the membrane. When the intracellular ascorbate reaches the level when Fe^3+^ is reduced to Fe^2+^ (assumed to be equal to 20 mM), the pool of labile iron increases, causing excessive ROS production leading to cell death (ferroptosis).

**Figure 2 cells-10-02964-f002:**
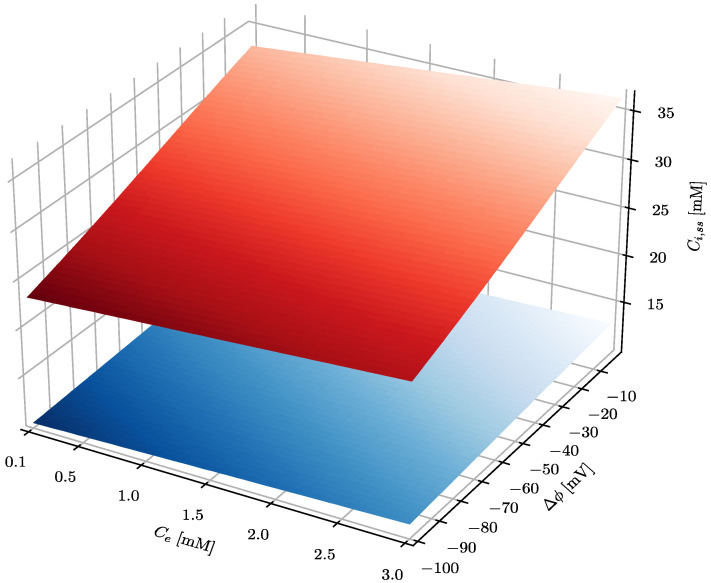
The effect of the plasma membrane electrical potential difference and ascorbate concentration in serum on the steady-state ascorbate concentration inside a cell. The serum pH_e_ equals 7.4, whereas intracellular pH_i_ equals 7.5 and 7 for cancer and healthy cells, respectively. The blue and red surfaces show the intracellular steady-state level of ascorbate in healthy and cancer cells, respectively.

**Figure 3 cells-10-02964-f003:**
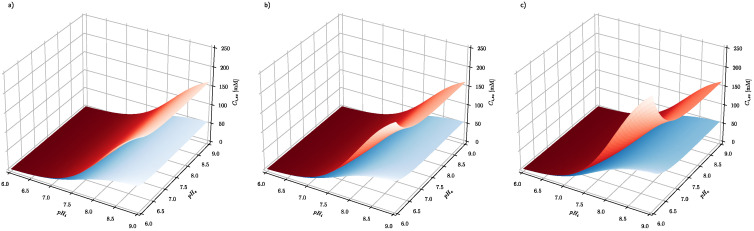
The effect of extracellular and intracellular pH on the intracellular steady-state ascorbate concentration. Calculations have been performed for system, where extracellular ascorbate concentration (**a**) C_e_ = 0.2 mM (**b**) C_e_ = 0.5 mM (**c**) C_e_ = 1 mM and membrane electrical potential equals Δϕ=−10 mV and Δϕ=−100 mV for cancer (red surface) and healthy cells (blue surface), respectively.

**Figure 4 cells-10-02964-f004:**
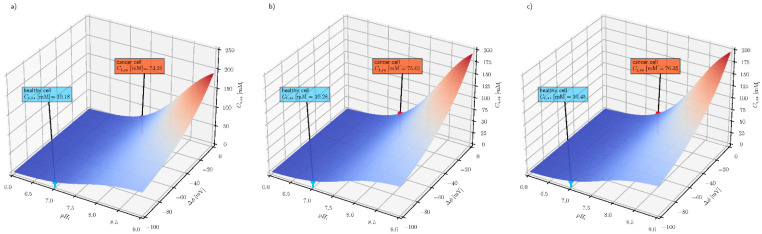
The effect of intracellular pH (pH_e_ = 7.4) and plasma membrane electrical potential difference on the steady-state intracellular ascorbate concentration for three extracellular ascorbate concentrations (**a**) C_e_ = 0.2 mM (**b**) C_e_ = 0.5 mM (**c**) C_e_ = 1 mM for cancer (red label) and health cells (blue label), respectively.

**Figure 5 cells-10-02964-f005:**
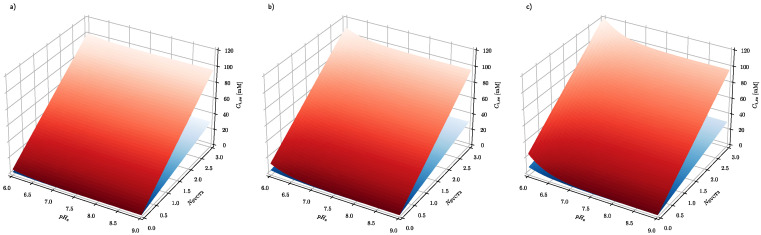
The effect of number of SVCT transporters and extracellular pH on the intracellular steady-state ascorbate concentration for cancer and healthy cells when exposed to (**a**) 0.2 mM (**b**) 0.5 mM (**c**) 1 mM extracellular ascorbate concentrations. Membrane electrical potential and pH equals $\Delta \phi =-10\;\mathrm{mV},\;pH_{i}=7.5$ and $\Delta \phi =-100\;\mathrm{mV},\;pH_{i}=7.0$ for cancer (red surface) and healthy cells (blue surface), respectively. The N_SVCT2_ is a number expressed in relative values, where 1 stands for the flux of ascorbate facilitated by transporters expressed in Xenopus oocytes cells [38].

**Figure 6 cells-10-02964-f006:**
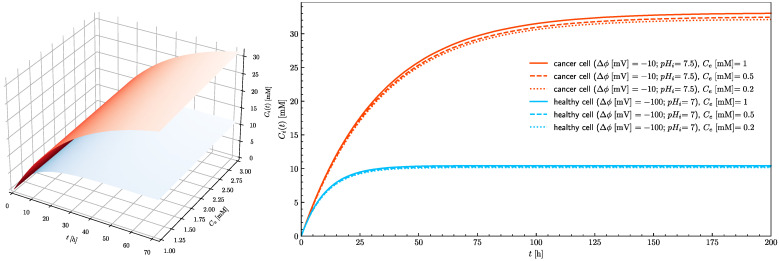
The effect of extracellular ascorbate concentration on the rise of the intracellular ascorbate concentration in time towards the steady-state value for healthy (blue) and cancer (red) cells (left panel). The right panel shows traces of intracellular ascorbate concentration as a function of time for healthy (blue traces) and cancer (red traces) cells. Traces show cases when extracellular ascorbate concentration equals 0.2, 0.5 and 1 mM for pH_e_ = 7.4.

**Figure 7 cells-10-02964-f007:**
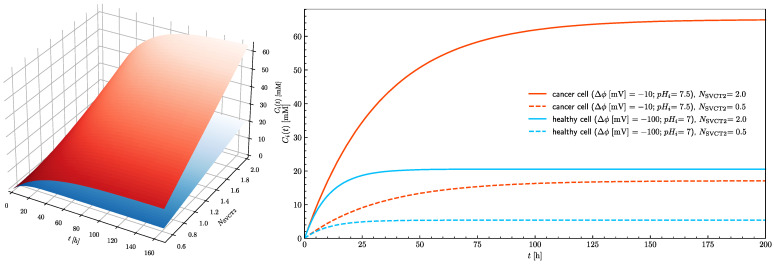
The effect of the number of SVCT2 transporters on the rise of the intracellular ascorbate concentration in time towards the steady-state value for healthy (blue) and cancer (red) cells (left panel). The right panel shows traces of intracellular ascorbate concentration as a function of time for healthy (blue traces) and cancer (red traces) cells. Traces show cases when the number of SVCT2 transporters equal 0.5 and 2 for pH_e_ = 7.4.

**Table 1 cells-10-02964-t001:** The numerical values of relevant parameters of healthy and cancer cells used for the calculations [22,24,25,26,27,28,29].

	Symbol	Value	Unit	Reference
Faraday constant	*F*	96,485	C/mol	
Gas constant	*R*	8.31	J/(mol·K)	
Temperature	*T*	310	K	
*pK_a_* of ascorbic acid	*pK_a_*	4.70		[24]
Membrane permeability of monoionic ascorbate	*P_i_*	3 × 10^−14^	m/s	[25]
Membrane permeability of neutral ascorbate	*P_n_*	1.1 × 10^−12^	m/s	[25]
Maximal rate of the active transport	*V* _max_	2.68 × 10^−4^	mM/s	[26]
Membrane electrical potential in cancer cell	Δ*φ*	−10	mV	[22]
Membrane electrical potential in healthy cell	Δ*φ*	−100	mV	[22]
Intracellular pH in healthy cell	*pH_i_*	7.0–7.2		[27]
Intracellular pH in cancer cell	*pH_i_*	7.12–7.65		[27]
Extracellular pH	*pH_e_*	7.35–7.45		
Intracellular metabolic rate	*v* _met_	4 × 10^−11^	Mol/(*s·g*)	[28]
Volume of eukaryotic cell	*V* _cell_	3.38 × 10^−15^	m^3^	[29]
Surface area of eukaryotic cell	*A* _cell_	1.35 × 10^−7^	m^2^	[29]
Mass of proteins in eukaryotic cell	*m* _protein_	6.75 × 10^−13^	g	[29]

## Data Availability

Not applicable.

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
