# Peer review of "The Membrane Electrical Potential and Intracellular pH as Factors Influencing Intracellular Ascorbate Concentration and Their Role in Cancer Treatment"

_cells, 2021, doi:10.3390/cells10112964_

Round 1
Reviewer 1 Report
The work is solid, important, and advances the field, therefore I recommend that the autors include conclusion in the manuscript, although the conclusion is not obligatory according to the instructions of the authors in the preparation of the manuscript. It would be very useful to include in the manuscript a conclusion in which the results will be critically summarized.
Author Response
Point 1. The work is solid, important, and advances the field, therefore I recommend that the authors include conclusion in the manuscript, although the conclusion is not obligatory according to the instructions of the authors in the preparation of the manuscript. It would be very useful to include in the manuscript a conclusion in which the results will be critically summarized.
Response 1: Following reviewer’s advice, we added Conclusions at the end of the manuscript in the following form:
Conclusions
The concentration of ascorbate inside a cell is a result of three processes, metabolic degradation, active transport facilitated by dedicated transporters (SVCT1 and SVCT2) and the passive diffusion across the lipid bilayer of the plasma membrane. The dependence of the passive transport on ascorbate protonation and plasma membrane potential difference can be used for selective targeting of ascorbate to cancer cells. Specifically, most cells constituting the human body are highly differentiated. Such cells are characterized by, among other things, high negative inside plasma membrane electrical potential (reaching 100 mV). Cancer cells are different, their electrical plasma membrane potential is usually low and does not exceed 10 mV. This difference can be used to elevate the ascorbate concentration in the cytoplasm of cancer cells without affecting healthy cells. This effect may justify the application of ascorbate as the supportive treatment in cancer therapy characterized by none or low systemic toxicity. Simulations presented in the paper show that such treatment to be effective requires maintenance sufficient, albeit not necessary excessively high, ascorbate concentration in plasm and interstitial fluids. However, due to the limited capacity of the active transport, the ascorbate exposure should extend in time for days rather than hours.
We also added some comments in the discussion section to stress the model conclusions.
We would like to appreciate the reviewer comments, which not only improve the manuscript quality, but also shows the possible avenues for further experimental and theoretical studies.
We hope that we satisfactory addressed issue raised by reviewer and that the manuscript in current form is suitable for publication.
Best regards,
Paulina Dałek
Reviewer 2 Report
In general, it was difficult to properly evaluate this paper without feedback from the author. The theoretical framework upon which the equations (and subsequent figures) were constructed has a number of incorrect assumptions (see below). It is difficult to tell which of these assumptions are critical to the findings. In its current state, the manuscript seems overly simplistic and not supported by clinical findings.
Concerns:
- In the introduction, the authors suggest that cancer cell death via ascorbate occurs through ferroptosis. While this is no doubt the case for cell culture, and to some degree in xenografts, no study has looked at the iron-mediated cell death in cancer cells in situ. Cell culture is a terrible place to study ascorbate and iron (for various reasons). Therefore, the comments on lines 77-81 should be softened accordingly.
- Line 89: Cells do not typically accumulate DHA. DHA is rapidly reduced to ascorbate or undergoes spontaneous ring cleavage to 2,3-DKG and smaller carbohydrates.
- Related to #2, on Line 90-92: Cells do not typically accumulate excess ascorbate. Regulation of the SVCT proteins on the cell membrane prevents this from occurring under physiological conditions and activation of organic anion transport proteins can efflux ascorbate if concentrations get too high. There are also possible efflux pathways through the ER. References 23 and 24 here do not appear to involve intact cells to demonstrate these processes.
- Lines 121-124: What is the initial assumed concentration of ascorbate inside the cells? What are the assumptions related to labile iron in these cells? Also, what published evidence are these assumptions based on? As stated before, cell culture studies are inadequate for these values.
- Line 131: Is Jcell(t) assumed to be linear? This is not the case in vivo.
- Line 141-144: Intracellular concentration is not the only variable involved in the cytotoxicity of ascorbate. As seen, multiple cancer cells types are resistant to ascorbate despite high delivery - they behave like normal cells. You cannot assume that cancer cells have a large labile iron pool a priori.
- Line 144-145: This is making the assumption that cytotoxic effects of ascorbate can only occur inside the cell. This is contrary to several lines of evidence that suggests otherwise – that ascorbate has a pro-oxidant effect in interstitial fluids.
- Line 149-150: This range of intracellular ascorbate levels is not available to all tissues. Most organs in the body will never achieve 10 mM ascorbate.
- Line 162: Ascorbate concentrations in the body hardly ever will exceed 100 mM, even with massive doses of ascorbate administered by IV
- Line 166: This is a simplistic view of active ascorbate accumulation into cells. First, the number of available SVCT proteins on the cell surface is variable - they respond to conditions. Second, SVCT1 and SVCT2 can be present in some cell types. Third, GLUT transport proteins also contribute to ascorbate accumulation via DHA transport and reduction.
- Line 188: It is unclear how vmet was determined. Neither reference provides this number. Is vmet the consumption of ascorbate directly equal to the consumption of glucose? It's unclear how this relationship can be assumed.
- Figure 2: Are there any literature sources to support ascorbate concentrations being dramatically higher in cancer tissues following ascorbate treatment?
- Figure 3: It does not appear that the ionization state of ascorbate is contributing to the effects of ascorbate at different pH, when we know the opposite is true - especially below pH 7 and above pH 8
Author Response
In general, it was difficult to properly evaluate this paper without feedback from the author. The theoretical framework upon which the equations (and subsequent figures) were constructed has a number of incorrect assumptions (see below). It is difficult to tell which of these assumptions are critical to the findings. In its current state, the manuscript seems overly simplistic and not supported by clinical findings.
Concerns:
Point 1: In the introduction, the authors suggest that cancer cell death via ascorbate occurs through ferroptosis. While this is no doubt the case for cell culture, and to some degree in xenografts, no study has looked at the iron-mediated cell death in cancer cells in situ. Cell culture is a terrible place to study ascorbate and iron (for various reasons). Therefore, the comments on lines 77-81 should be softened accordingly.
Response 1: We agree with arguments stated by the reviewer. The text is modified accordingly
Point 2: Line 89: Cells do not typically accumulate DHA. DHA is rapidly reduced to ascorbate or undergoes spontaneous ring cleavage to 2,3-DKG and smaller carbohydrates.
Response 2: We appreciate the reviewer comment. The text is changed accordingly.
Point 3: Related to #2, on Line 90-92: Cells do not typically accumulate excess ascorbate. Regulation of the SVCT proteins on the cell membrane prevents this from occurring under physiological conditions and activation of organic anion transport proteins can efflux ascorbate if concentrations get too high. There are also possible efflux pathways through the ER. References 23 and 24 here do not appear to involve intact cells to demonstrate these processes.
Response 3: We appreciate the comment. The unfortunate term “accumulation” was removed, and the statement reformulated.
The epigenetic response of cancer cells to elevated ascorbate concentration is discussed later in the text. It has been demonstrated that indeed the changed number of transporters significantly alters the intracellular ascorbate concentration. We softened the questioned statement and removed the two indicated references.
Point 4: Lines 121-124: What is the initial assumed concentration of ascorbate inside the cells? What are the assumptions related to labile iron in these cells? Also, what published evidence are these assumptions based on? As stated before, cell culture studies are inadequate for these values.
Response 4: It has been assumed that both; initial ascorbate concentration and labile iron level at the beginning of the simulation equal zero. These values will not affect the calculated steady-state ascorbate levels inside cells. They will affect however the kinetics of the ascorbate rise in relevant cytoplasm.
Due to the high metabolic activity of cancer cells, they will have depressed ascorbate levels, as frequently observed in cancer patients. Therefore, it can be assumed that such an assumption is not dramatically incorrect. It has been also demonstrated that iron homeostasis in tumours and cancer cells are also altered [1][2]. Again, the presence of labile iron in the extracellular space will contribute to the ROS level and ascorbate degradation. The latter can be accounted for in the elimination term. We agree with the reviewer that such concerns are valid when the ascorbate effect on cancer cells inside a tumour is investigated. The presented analysis was reduced to the simplest possible scenario to demonstrate that ascorbate accumulation inside the cell is different from exogenous active substances due to the existence of dedicated and metabolically supported active transport. In addition, the modification of the parameters describing extracellular fluid has to address a specific type of tumour accounting for its morphology and properties of the surrounding tissues, which would likely reduce the generality of the analysis.
Point 5: Line 131: Is Jcell(t) assumed to be linear? This is not the case in vivo.
Response 5: On the physiological level, the term Jcell(t) is assigned to the ascorbate intake by cells.t In such case he flow is not a linear function of time due to the saturated character of SVCT transporters. It depends on cellular deficiency as well as temporal metabolic activity requirements.
Point 6: Line 141-144: Intracellular concentration is not the only variable involved in the cytotoxicity of ascorbate. As seen, multiple cancer cells types are resistant to ascorbate despite high delivery - they behave like normal cells. You cannot assume that cancer cells have a large labile iron pool a priori.
Response 6: The reviewer raises the valid point that there are differences between cancer cells. In the manuscript we simplify the model by selecting the ferroptosis as a therapeutic target. Differences between cancer cells are not accounted for in the current version of the manuscript. The model can be adopted for a specific case if required. We change the text so the therapeutic effect in the form of rising iron labile pool caused by the elevated ascorbate concentration is clearly stated. We did not assume that there is a large pool of labile iron in cancer cells before treatment.
Point 7: Line 144-145: This is making the assumption that cytotoxic effects of ascorbate can only occur inside the cell. This is contrary to several lines of evidence that suggests otherwise – that ascorbate has a pro-oxidant effect in interstitial fluids.
Response 7: We agree with the argument stated by the reviewer that ascorbate can inflict the toxic effect in interstitial fluids. However, the effect depends on the local environment and will likely synergize with the process address in the paper. Again, such issue can be address when analysing a specific type of cancer.
Point 8: Line 149-150: This range of intracellular ascorbate levels is not available to all tissues. Most organs in the body will never achieve 10 mM ascorbate.
Response 8: The statement is correct. It is why we mentioned the range of possible intracellular ascorbate concentrations from 0.05 mM to 10 mM.
Point 9: Line 162: Ascorbate concentrations in the body hardly ever will exceed 100 mM, even with massive doses of ascorbate administered by IV
Response 9: The reviewer correctly pointed out that the ascorbate concentration given is not correct. The value was corrected from 100 mM to 100 M.
Point 10: Line 166: This is a simplistic view of active ascorbate accumulation into cells. First, the number of available SVCT proteins on the cell surface is variable - they respond to conditions. Second, SVCT1 and SVCT2 can be present in some cell types. Third, GLUT transport proteins also contribute to ascorbate accumulation via DHA transport and reduction.
Response 10: We appreciate the reviewer comment since it shows the pathway, which can be taken future developing the model. However, we assumed, following the literature, that epithelial cells have both transporters whereas other cells have mainly SVCT2 transporter. For the current model, SVCT2 transporters are more relevant, since they have a higher affinity to ascorbate. It would be interesting to extend the model for cases where both transporters are present.
We assumed that the DHA does not contribute significantly to the flux of ascorbate as stated earlier. This is because the quantity of ascorbate does not exceed 1% of ascorbate inside cells. In addition, it is believed that DHA is reduced inside cells and/or flows out of the cell as demonstrated by homeostasis of ascorbate in the brain. We used these arguments to neglect the effect of DHA. It can also be argued that the effect of DHA is a part of the metabolic flux.
Point 11: Line 188: It is unclear how vmet was determined. Neither reference provides this number. Is vmet the consumption of ascorbate directly equal to the consumption of glucose? It's unclear how this relationship can be assumed.
Response 11: We provide an argument for the assumption, albite it is not based on the direct demonstration that ascorbate concentration in cancer cells is higher than in normal cells. Nevertheless, we add in the text a proposition for the correlation in the following form:
“The approximation is based on correlation between two observations; that metabolic activity of cancer cells exceeds that of normal cells and that the ascorbate required for sick patients to reach homeostatic level is much higher than for healthy persons [3, 4]”
Point 12: Figure 2: Are there any literature sources to support ascorbate concentrations being dramatically higher in cancer tissues following ascorbate treatment?
Response 12: The model assumed that ascorbate concentration is the same in all interstitial fluids and that differences between cells depends on passive and active transports. The active transport is epigenetically controlled, whereas the passive transport is affected by the level of cell differentiation and local aqueous phase properties. The demonstration that ascorbate concentration in cancer cells exceed that in healthy cells has not been provided yet. Perhaps the publication will stimulate such studies.
Point 13: Figure 3: It does not appear that the ionization state of ascorbate is contributing to the effects of ascorbate at different pH, when we know the opposite is true - especially below pH 7 and above pH 8.
Response 13: The extracellular and intracellular pH will mainly affect the passive transport, since the lipid bilayer permeability for protonated ascorbate is two orders of magnitude larger than for its mono-ionic form [5]. Consequently, the dynamic processes rather than stead-state level will be affected. Presented result shows that the plasma membrane potential is a main factor affecting the steady-state ascorbate concentration in a cell, assuming of course that number of transporters will not change.
We would like to appreciate the reviewer comments, which not only improve the manuscript quality, but also shows the possible avenues for further experimental and theoretical studies.
We hope that we satisfactory addressed all issues raised by reviewer and that the manuscript in current form is suitable for publication.
Best regards,
Paulina Dałek
References
- Sacco, A., et al., Iron Metabolism in the Tumor Microenvironment-Implications for Anti-Cancer Immune Response. Cells, 2021. 10(2).
- Ying, J.F., et al., The role of iron homeostasis and iron-mediated ROS in cancer. Am J Cancer Res, 2021. 11(5): p. 1895-1912.
- Aditi, A. and D.Y. Graham, Vitamin C, gastritis, and gastric disease: a historical review and update. Dig Dis Sci, 2012. 57(10): p. 2504-15.
- Liu, M., et al., Vitamin C increases viral mimicry induced by 5-aza-2'-deoxycytidine. Proc Natl Acad Sci U S A, 2016. 113(37): p. 10238-44.
- Hannesschlaeger, C. and P. Pohl, Membrane Permeabilities of Ascorbic Acid and Ascorbate. Biomolecules, 2018. 8(3).